# Trumpet is an operating system for simple and robust cell-free biocomputing

Judee A. Sharon[1], Chelsea Dasrath[1], Aiden Fujiwara[2], Alessandro Snyder[2], Mace Blank[1], Sam O'Brien[2], Lauren M. Aufdembrink[1], Aaron E. Engelhart[1] & Katarzyna P. Adamala [1]✉

Biological computation is becoming a viable and fast-growing alternative to traditional electronic computing. Here we present a biocomputing technology called Trumpet: Transcriptional RNA Universal Multi-Purpose GatE PlaTform. Trumpet combines the simplicity and robustness of the simplest in vitro biocomputing methods, adding signal amplification and programmability, while avoiding common shortcomings of live cell-based biocomputing solutions. We have demonstrated the use of Trumpet to build all universal Boolean logic gates. We have also built a web-based platform for designing Trumpet gates and created a primitive processor by networking several gates as a proof-of-principle for future development. The Trumpet offers a change of paradigm in biocomputing, providing an efficient and easily programmable biological logic gate operating system.

The reality of biological computing hardware is closer than it has ever been. One of the most well-studied biological computing systems is the live cell logic gate. We now have many examples of engineered cells receiving inputs in the form of light[1–3] or chemical compounds[4], performing an internal calculation, and outputting a protein signal. Through the earlier work on genetic circuits[5–7] in bacteria and the more recent advances in tunable[8] and precisely edited[9] logical expression systems, we can picture a future of cellular devices that take advantage of complex bacterial and mammalian[10] genomes.

Another biocomputing system, in vitro enzyme-free DNA logic gates, also has a rapidly growing body of knowledge. Earlier work presented DNA as a code-able polymer that is much more adaptable to nanoscale electronics than silicon-based circuitry[11,12]. Now DNA strand displacement technologies have progressed to include reusable NAND gates[13] and methods for studying cell population behaviors through communication between non-lipid protocellular DNA logic gates[14].

While in vitro work using biological enzymes as catalytic machines for DNA-based molecular computing[15–17] was first published over two decades ago, this middle ground lagging behind live cell and enzyme-free work[18]. There is a need for further development of a computing technology that harnesses the evolutionary strengths of biological components (both DNA and enzymes), without including the complexity of genomes, endogenous live processes, or competition-based strand displacement methods. It is likely that the future of biocomputing hardware and software will likely include a combination of all three technologies[19]: live cell, enzyme-free, and enzymatic logic gates.

To address the need for the third method—an enzymatic and cell-free logic gate system—we present a new platform: the Transcriptional RNA Universal Multi-Purpose Gate Platform, or Trumpet. This biocomputing platform can process digital signals of DNA inputs in Boolean logic gates, followed by either DNA outputs or fluorescent RNA aptamer[20] outputs via cell-free transcription. Trumpet uses DNA as a polymer that acts both as the wires leading to the circuit and the circuit itself. The circuit employs restriction enzymes or polymerases for robust processing, utilizing nature's highest fidelity catalysts. After completion of the circuit, Trumpet uses the cell-free environment to transcribe the DNA into a fluorescent RNA aptamer, which in turn acts as the "lightbulb at the end of the circuit board". The use of transcription to produce an output provides signal amplification: each strand of DNA that comes out of the logic gate is a template for many strands of RNA aptamers, increasing the number of fluorescent molecules providing the readout.

[1]Department of Genetics, Cellular Biology, and Development, University of Minnesota, Twin Cities, Minneapolis, MN, USA. [2]Department of Computer Science, University of Minnesota, Twin Cities, Minneapolis, MN, USA. ✉e-mail: kadamala@umn.edu

Trumpet operations are performed in a relatively simple reaction environment, combining the benefits previously attributed to live cell logic gates (signal amplification, enzymatic multiple turnover) with the advantages of robust in vitro environments, like toehold strand displacement platforms.

We have validated the performance of Trumpet with all basic Boolean logic gates (NAND, NOT, NOR, AND, and OR), and we have demonstrated the operation of a multilayer processor constructed from a several universal gates. We also developed a web-based tool facilitating the design of sequences for the Trumpet platform.

## Results
### Preliminary testing

Restriction enzymes are integral to the operation of the Boolean logic gates designed in this study. These enzymes have largely been used in modern biology to clone and genetically manipulate DNA. However, the ability of restriction enzymes to recognize, bind, and cleave a specific set of DNA nucleotides is the characteristic we exploit for Boolean gate function. Type II restriction enzymes, like those used in the following experiments, often function as homodimers[21] where both subunits bind DNA non-specifically at first, then change the conformation of DNA at the recognition site prior to catalytic cleavage. The NAND, NOT, and NOR logic gates use a Type II restriction enzyme and a corresponding recognition sequence at the gate site, where the DNA may or may not be double-stranded depending on the presence of inputs. When there is a lack of inputs, the DNA template is single-stranded, which potentially prevents the restriction enzyme from conformationally changing the DNA to an extent necessary for cleavage. The interaction of single- versus double-stranded DNA and the enzyme is a major facet of gate function. It is also important that all the enzymes necessary for various gate reactions—restriction enzyme, DNA polymerase, and RNA polymerase—function correctly in a single-buffered system for ease of use. Although restriction enzymes are required for the NAND, NOT and NOR gates, AND and OR gates are contingent on DNA polymerase, and RNA polymerase is necessary for the cell-free transcription of all gates.

Using New England BioLab's NEBuffer Activity/Performance Chart with Restriction Enzymes, several enzymes were chosen according to their continued activity in temperatures over 90 °C (i.e., no heat inactivation), short incubation periods, recognition sites without ambiguous bases, longer recognition sites to aid with specificity, digestions within recognition sites (instead of downstream of the sites), and activity in OneTaq DNA Polymerase Buffer (Supplementary Data 1 and SI Fig. 1). Eight restriction enzymes that satisfied all parameters were used in the initial tests for validating the obligatory double-stranded template requirement. The requirement for the recognition site to be double-stranded is a key property of the logic gate platform—it is only when the inputs are hybridized with the gate template that the template should be digested by the restriction enzyme.

The early digestion tests were conducted using a custom-made buffer, aHOT 7.9, that supports restriction enzyme digests, DNA polymerase reactions, and cell-free transcriptions. aHOT 7.9 contains the reagents found in the NEB OneTaq DNA Polymerase buffer, but also contains spermidine and dithiothreitol (DTT) and is buffered to pH 7.9 to aid in cell-free transcription[22,23].

Cell-free transcription and translation systems are a model for recapitulating endogenous cell processes (i.e., transcription of DNA and translation of RNA to proteins) in a modular, bottom-up fashion. By adding specific DNA templates and finite concentrations of small molecules, like ATP, NTPs, and amino acids, we can study how minute changes in the template affect downstream expression of protein[24]. In this study, however, we focus on the transcription as the signal amplification mechanism, rather than translation to avoid further increasing the complexity in the processivity of the logic gates.

### The NAND gates

Four restriction enzymes, PvuII, BsaAI, NruI, and RsaI, were found to digest only double-stranded DNA templates and were also functional in aHOT 7.9 (SI Fig. 2). These restriction enzymes were validated through an early design of the NAND gate. The NAND gate is composed of a single-stranded 105-nucleotide gate template, two single-stranded 15-base inputs that are complementary to regions on the gate template, and a single-stranded T7 Max RNA Polymerase promoter complementary to another region on the gate template. The gate template is an antisense strand containing the T7 Max promoter[25], a random sequence of 12-bases, a 6-base restriction enzyme recognition site, another random sequence of 12-bases, and the DNA sequence of an RNA aptamer (SI Fig. 3). Each input is the sense complement for one set of 12 random bases and 3 bases of the restriction enzyme recognition site. To operate the gate, the minimum components of a gate template, the T7 Max sense strand, and the restriction enzyme in aHOT 7.9 are required. When both inputs are provided, each input hybridizes with the complementary regions on the gate template, making the restriction enzyme recognition site double-stranded (Fig. 1a). The restriction enzyme can recognize and cut the gate at the recognition site. The promoter and aptamer sequences are no longer one contiguous sequence. When the gate is processed by RNA polymerase in a cell-free transcription, the RNA aptamer sequence is not transcribed, and no fluorescence is detected. The lack of fluorescence in the NAND gate with both inputs is recorded as a 0 signal. When zero or one input is added, the recognition site on the gate template remains single-stranded. The restriction enzyme cannot digest a wholly or partially single-stranded recognition site, so the gate template remains intact. Since the promoter and aptamer regions remain connected, RNA polymerase can transcribe the entire template, producing the fluorescent RNA aptamer. The fluorescent signal of a NAND gate with zero or one input is recorded as a 1 (Fig. 1b, c).

In first-round experiments with all gates, we used Broccoli[26,27] as the RNA aptamer (SI Fig. 4). Assuming that transcribed Broccoli can fold into its correct secondary structure, it binds and activates the fluorescence of the ligand DFHBI (4-[(3,5-difluoro-4-hydroxyphenyl)methylidene]−1,2-dimethyl-4,5-dihydro-1H-imidazol-5-one). To demonstrate versatility in RNA aptamer choice, Fig. 1d–k show NAND gate signal outputs through Pepper[28], Mango[29], Corn[30], and Malachite Green[31] aptamers.

All four selected restriction enzymes produced a NAND gate signal pattern as expected. PvuII[32,33] produced the best difference in signal between 1 and 0 and performed the most efficiently in aHOT 7.9 (SI Fig. 2a). Figure 1c shows that the 1 output is 90 times higher on average than the 0 output. In most cases, unless explicitly stated otherwise, input concentrations (6 μM) were ~3× higher than the gate template concentrations (2 μM) when PvuII is used as the restriction enzyme.

During the experiments testing different restriction enzymes, we confirmed that the concentration ratios of gate template to inputs did not significantly affect the output signal (SI Fig. 5). Likewise, we also established that the concentration differences between the restriction enzymes and the gate templates were not significantly affecting the gate performance within the tested range. When inputs were present, restriction enzyme concentrations between 5 Units and 40 Units were not found to make a significant difference in the 0 signal (SI Fig. 6). All enzyme concentrations enabled accurate gate operation.

NAND gate reactions are also successful at gate template and input concentrations that are 0.05× of the standard concentrations used in many of the experiments in this paper (SI Fig. 7). Our chosen standard concentration of 2 μM gate template and 6 μM inputs provides reliable 1 and 0 signal. However, we see comparable signal differences when 1 μM gate template and 3 μM inputs were used. There was a slight decrease in signal when 0.5 μM gate template and 1.5 μM inputs were used. From the samples were 100 nM gate template and 300 nM inputs were used, there was a sharp drop off in overall

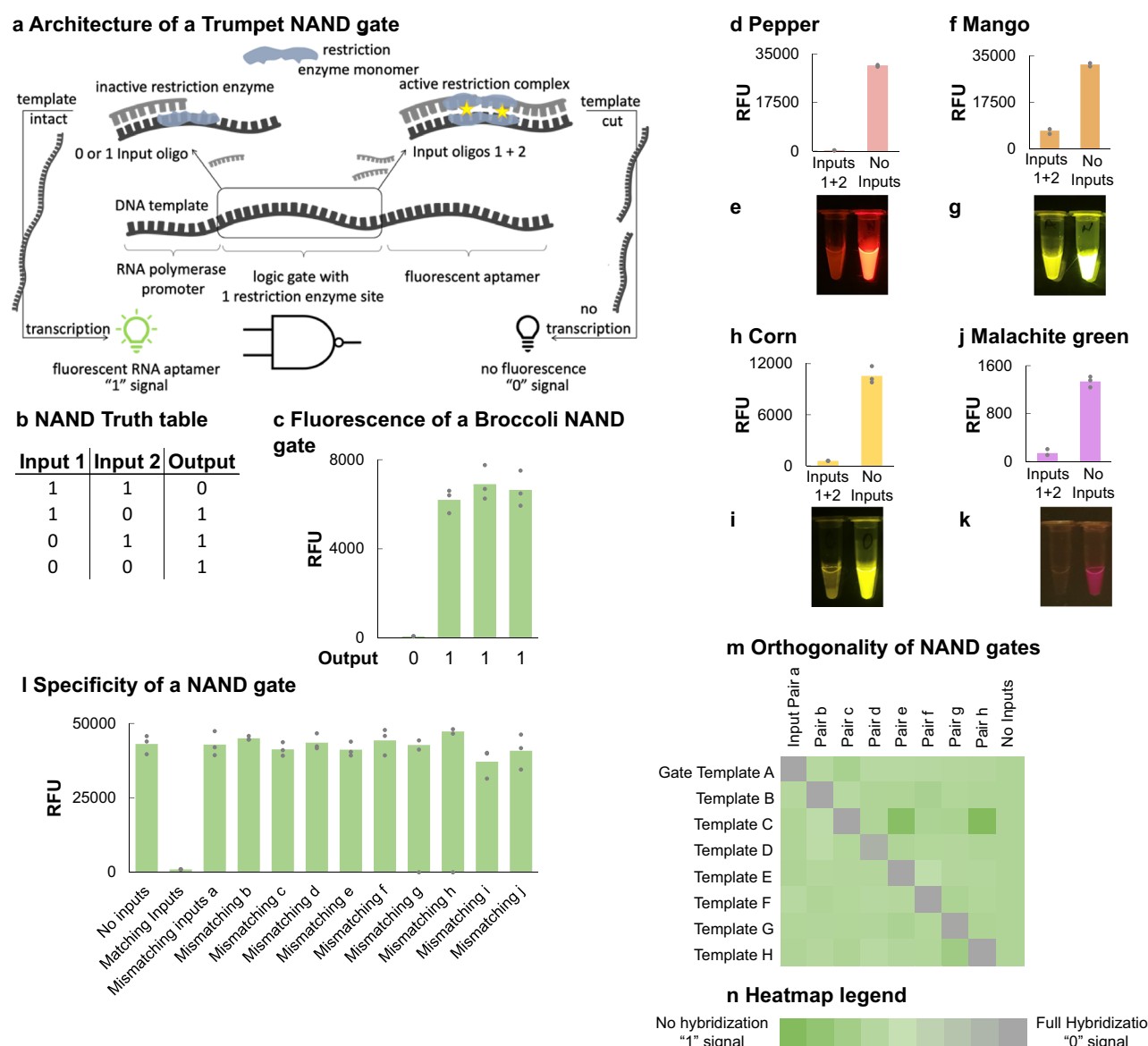

**Fig. 1 | Trumpet NAND gates. a** The general architecture of a NAND gate. **b** A typical NAND gate truth table. **c** Fluorescence results of a NAND gate. When no inputs, Input 1, or Input 2 hybridize with the gate template, the template remains intact and the results are a 1 when Broccoli, the encoded RNA aptamer, is transcribed. When both inputs hybridize with the gate template simultaneously, the template is cut by the restriction enzyme and Broccoli is not transcribed. This results in a 0, a lack of signal. **d** Fluorescence results of a NAND gate transcribing Pepper aptamer. **e** Visual results of the signals shown in (**d**). **f** Fluorescence results of a NAND gate transcribing Mango aptamer, which binds TO1-PEG-biotin as the ligand. **g** Visual results of the signals shown in (**f**). **h** Fluorescence results of a NAND gate transcribing Corn aptamer, which binds DFHO as the ligand. **i** Visual results of the signals shown in (**h**). **j** Fluorescence results of a NAND gate transcribing Malachite Green aptamer, which binds Malachite Green Ligand. **k** Visual results of the

signals shown in (**j**). **l** Fluorescence results showing specificity of inputs to their gate templates. Incorrect pairings are comparable with the results of the gate when no inputs are added, and the gate template remains intact. In contrast, when the correct pair of inputs is mixed with the gate template, digestion occurs, preventing the transcription of Broccoli, shown in the sample labeled "Matching Inputs". **m** Heatmap showing input specificity for eight unique NAND gate templates. **n** Heatmap legend showing that green squares on the heatmap represent 0% mismatch or inputs that are a direct match to a gate template and can hybridize correctly. The dark gray squares represent 100% mismatch, or inputs that are not a match to a gate template and will not hybridize. On all panels, the value of each replicate within a sample set is represented by a gray marker. The green bars are the averages of each sample set ($n = 3$, each experiment was repeated three times). Source data are provided as a Source Data file.

detected fluorescence but the reactions without inputs are still produce a 1 signal that is at least 2 times greater than the reactions with both inputs. The NAND gate was still operational with gate template concentrations of 10 nM. The steep drop-off in signal at the nanomolar concentrations could be due to fewer DNA molecules that are able to encounter the matching inputs for hybridization or are able to bind with the restriction enzymes for digestion.

The length of each input (15 bases) and the randomness of the 12 bases flanking each restriction enzyme cut site ensures gate template specificity. Figure 1l shows that only the matching input pair can

hybridize with a corresponding gate template, and result in a 0 signal. All other mismatching inputs are unable to hybridize, and the result is a 1. The gates have the potential for operating in a highly orthogonal manner with minimal cross talk, especially in multiplexed reactions. The heatmap in Fig. 1m further shows that pairs of inputs are optimized specifically to match with gate templates. The heatmap legend in Fig. 1n indicates that gate templates with mismatching input pairs will not hybridize well and will result in Broccoli transcription and fluorescence (green). In contrast, the gate templates with the correct, matching inputs will hybridize, be cleaved, and will not result in

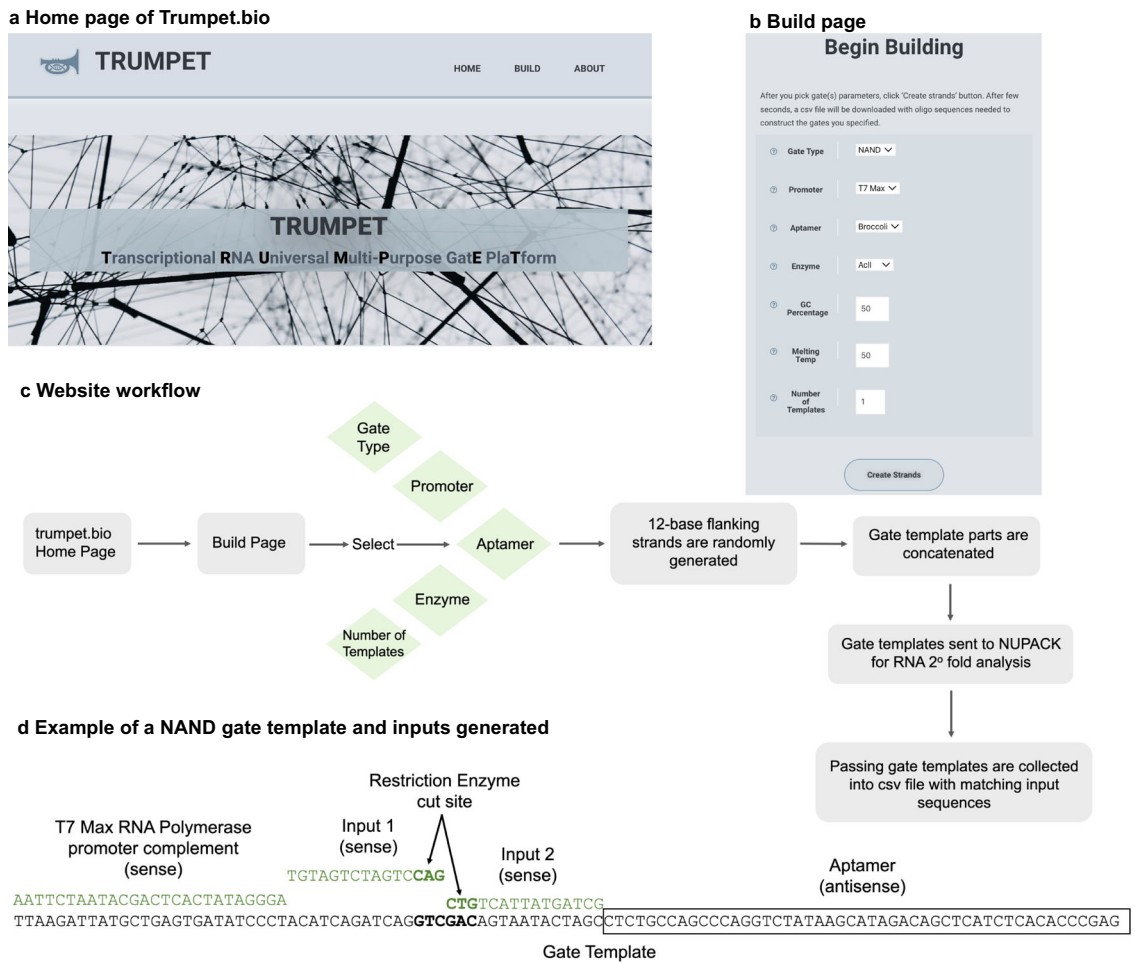

**Fig. 2 | Trumpet web platform. a** The Home page of the Trumpet.bio design platform. **b** The Build page showing the selection options−Gate Type, Promoter, Reporter, Enzyme, GC Percentage, Melting Temp, and Number of Strands. **c** Platform workflow from Home page to designed-strand output. **d** An example of a gate sequence built with Trumpet. The antisense (bottom) strand is the gate template. The top strands are the complementary T7 Max promoter sequence or the gate inputs.

transcribed Broccoli (gray). Individual analysis for each gate template-input combination can be found in Supplementary Data file 2 and the corresponding spreadsheet.

## Trumpet design platform

The semi-rational design of the gate (i.e., the random flanking bases combined with a specific and consistent restriction enzyme recognition sequence) is a time-saving measure that lends to the orthogonality of the platform. If the flanking bases were designed rationally, with thought to each neighboring nucleotide, each gate could take much longer to design as a whole. With this semi-rational approach, the manual design of each gate takes ~10 min. Predictive folding algorithms, like NUPACK and mFold, were used to guarantee that the RNA aptamer in the gate output would be able to fold into the correct secondary structure without interference from the upstream gate regions that would also be transcribed. Designing gates manually for high throughput reactions will take many hours, even employing semi-rational design. The Trumpet, **T**ranscriptional **R**NA **U**niversal **M**ulti-**P**urpose Ga**t**e **P**latform, design tool (Fig. 2a, b) was developed to address the time-intensive nature of high throughput gate design for this platform. When designing each gate template, a user defines each aspect on the gate template and Trumpet concatenates the T7 Max promoter sequence, the gate region, and the DNA sequence of the RNA aptamer. The tool designs the gate region by randomly assigning 12

bases to flank each side of the recognition site of a user-defined restriction enzyme. Then two inputs are designed by splitting the cut site in half and taking the complements of each side of the gate. Each input is the sense complement to the antisense gate region and will each contain the complement of the 12 random bases and the 3 bases of half of the restriction enzyme cut site. Trumpet repeats this action for every gate template requested, and outputs a csv file containing the antisense gate template sequences and the corresponding sense input sequences (Fig. 2c, d). 96 NAND gates can be designed by Trumpet in 15 s, compared to the ~16 h required for manual design of an equivalent number of gates.

## The NOT and NOR gates

The NOT and NOR gates follow a similar architecture to the NAND gate. The NOT gate template is 101 bases and composed of the antisense T7 Max promoter sequence, 10 random bases, a 6-base restriction enzyme cut site, another 10 random bases, and the antisense DNA sequence of an RNA aptamer (SI Fig. 8). This gate only requires one input, in accordance with its truth table, which is a 26-based sense sequence complementary to the entire random base and cut site region (Fig. 3a). The other minimum components of the gate: the T7 Max sense complementary sequence, and a restriction enzyme, are required for gate operation. When the input is present, it hybridizes with the gate region on the template, making the restriction enzyme

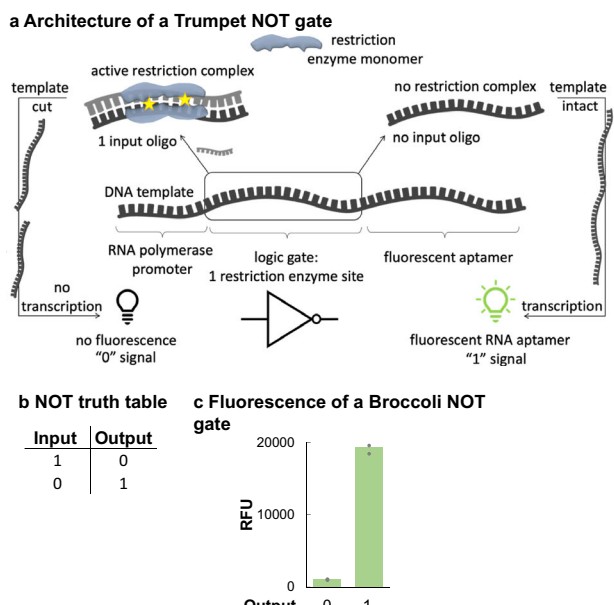

**a Architecture of a Trumpet NOT gate**

**b NOT truth table**

| Input | Output |
|-------|--------|
| 1 | 0 |
| 0 | 1 |

**c Fluorescence of a Broccoli NOT gate**

**d Architecture of a Trumpet NOR gate**

**e NOR truth table**

| Input 1 | Input 2 | Output |
|---------|---------|--------|
| 1 | 1 | 0 |
| 1 | 0 | 0 |
| 0 | 1 | 0 |
| 0 | 0 | 1 |

**f Fluorescence results of a Broccoli NOR gate**

**Fig. 3 | Trumpet NOT and NOR gates. a** The general architecture of a NOT gate. The NOT gate only requires one input, a 26-base single-stranded DNA that is complementary to the region with the restriction enzyme cut site on the gate template. When the input is present (left), it hybridizes with the gate template. The restriction enzyme recognizes the cut site and cleaves the gate template, separating the promoter from the aptamer. The T7 RNA polymerase cannot transcribe the aptamer, and the lack of fluorescence is recorded as a 0 signal. When the input is not present (right), the restriction enzyme cannot cut the single-stranded template. The T7 can transcribe the RNA aptamer from the intact gate template, which results in a fluorescence signal recorded as a 1. **b** A NOT gate truth table. **c** Fluorescent results of a NOT gate encoding a Broccoli aptamer. **d** The general architecture of a NOR gate. A 127-base gate template contains a promoter, first 26-base digest region (10-base random sequence, a restriction enzyme cut site, another 10-base random sequence), a second 26-base digest region (another two random sequences and the same cut site), and ends with the antisense sequence of an RNA aptamer. In this gate, one input consists of one 26-base digest region. When either or both inputs are present with the gate template, hybridization occurs and the restriction enzyme can digest the gate template in one or both locations, separating the promoter from the aptamer sequence. This template cannot be transcribed, and the lack of fluorescence is recorded as a 0 signal. It is only when neither input is present that the gate template remains intact, and transcription of the RNA aptamer occurs. The fluorescence of the transcribed aptamer is recorded as a 1 signal. **e** A NOR gate truth table. **f** Fluorescent results of a NOR gate. On all panels, the value of each replicate within a sample set is represented by a gray marker. The green bars are the averages of each sample set ($n = 3$). All experiments were repeated three times independently. Source data are provided as a Source Data file.

cut site double-stranded. The restriction enzyme recognizes the cut site and cleaves the gate template. The RNA polymerase, in the cell-free transcription reaction, cannot transcribe the aptamer sequence because it is no longer attached to the rest of the gate template. The lack of fluorescence is recorded as a 0. When the input is not present, the restriction enzyme cut site remains single-stranded and the restriction enzyme is unable to cut the gate template. RNA polymerase transcribes the attached aptamer sequence and the subsequent fluorescent signal from the RNA aptamer is recorded as a 1 (Fig. 3b, c).

The NOR gate template is 127 bases and contains two separate regions of random bases and restriction enzyme cut sites. In between the antisense T7 Max promoter and the RNA aptamer sequences, the NOR gate is composed of two consecutive sets of 10 random bases, a restriction enzyme cut site, and another 10 bases, for a total of 52 bases (SI Fig. 9). The same restriction enzyme and cut site sequence is used for both sets of gate regions. When either of the inputs or both inputs are present, they hybridize to their respective complementary regions on the gate template. Each region contains a restriction enzyme cut site, so the restriction enzyme can cleave each region independently regardless of whether the other input is present (Fig. 3d). This concept adheres to the NOR truth table (Fig. 3e). When either or both inputs are present, RNA polymerase cannot transcribe the aptamer sequence, because it will be cleaved away from the rest of the template. The lack of fluorescence is recorded as a 0. Only when neither input is present the gate template remains intact, and RNA polymerase transcribes the whole template including the aptamer. RNA aptamer fluorescence, in this case, is recorded as a 1 for the NOR gate (Fig. 3f).

While these results of the NOR gate (Fig. 3f) show that the 1 signal is 440 times greater than the 0 signal, the signal-to-noise ratio (i.e., signal difference between 1 and 0) is 19 to 1 for the NOT gate (Fig. 3c). This could be due to differences in DNA binding thermodynamics between certain designs of gate templates and their matching inputs. While we did not specifically explore this possibility in this study, we saw similar phenomena within the NAND gate template designs used for the crosstalk experiments for the heatmap in Fig. 1m (Supplementary Data 2). Nucleotide-level kinetics have been found to play a role in DNA strand displacement (DSD) template design, a platform commonly used for molecular computing[34]. Unlike in DSD, where each strand can be rationally designed and base-level kinetics can be mitigated, rational design of each gate template in this platform would be too labor intensive, limiting the number of gates that can be created within a reasonable time frame. Using random bases in combination with known restriction enzyme recognition sites in the gates is the compromise between rationally designing each base in the gate template and generating hundreds of sequences in a short time frame despite variations in fluorescence outcomes.

## The AND and OR gates

The AND and OR gates follow a slightly different architecture by using DNA polymerases—rather than restriction enzymes—that interact with each gate, but still rely on cell-free transcription for signal output. The AND gate is a 105-nucleotide DNA sequence that starts with a T7 Max promoter, a 30-base random sequence, and ends with the sequence of an RNA aptamer (SI Fig. 10). Unlike the NAND, NOT, and NOR gates, there is no starting gate template. Instead, Input 1 is the sense strand from the beginning (Base 1) of the T7 Max promoter to the Base 76, which lies in the RNA aptamer. Input 2 is the antisense strand from Base

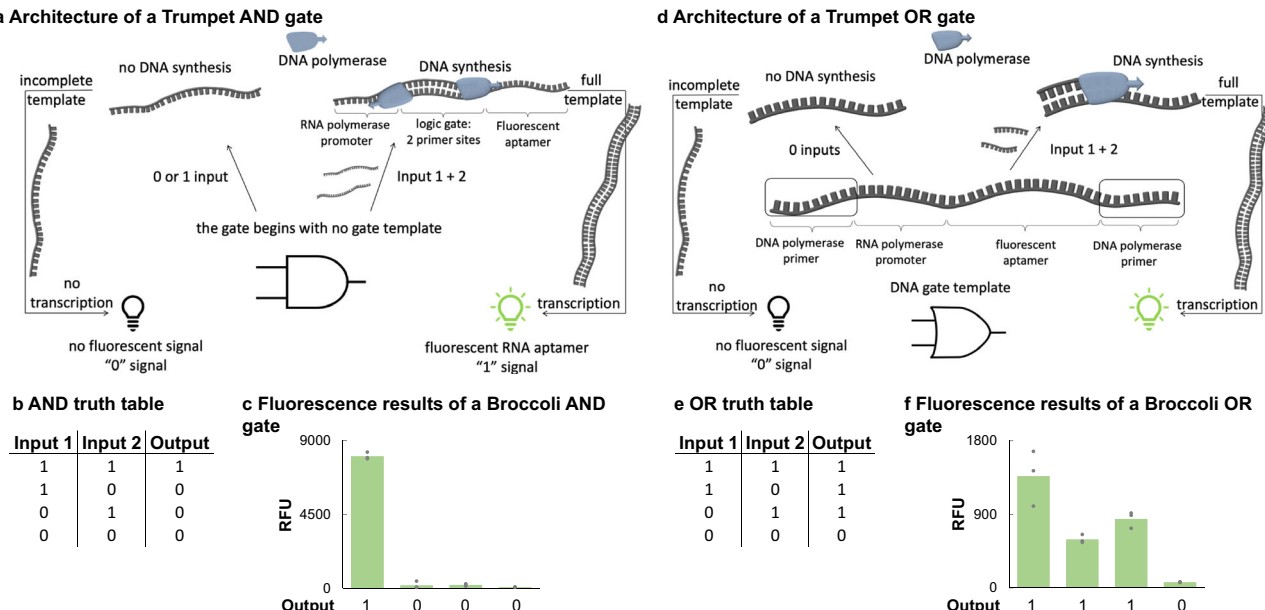

**Fig. 4 | Trumpet AND and OR gates. a** The general architecture of an AND gate. The gate begins without a gate template. Instead, the inputs are meant to hybridize directly with each other. A complete 106-base AND gate contains a T7 Max promoter sequence, a 30-base random sequence, and ends with the antisense sequence of an RNA aptamer. When zero or one input is present, the DNA polymerase is unable to extend the input to create a double-stranded gate. The T7 RNA polymerase relies on a double-stranded promoter sequence. This prevents the RNA polymerase from transcribing the aptamer, resulting in a 0 signal. When both inputs are present, DNA polymerase is able to extend each end to create a full double-stranded gate template. T7 RNA polymerase is then able to transcribe the RNA aptamer, and the aptamer fluorescent signal is recorded as a 1. **b** An AND gate truth table. **c** Fluorescent results of an AND gate encoding a Broccoli aptamer. **d** The general architecture of an OR gate. The gate template includes both 75-base

sense and antisense strands and encodes just the T7 Max promoter sequence followed by an RNA aptamer sequence. When the sense version of the gate template is present with Input 2, DNA polymerase can extend the template in one cycle to create a double-stranded sequence. Likewise, the antisense version of the gate template can be extended when paired with Input 1. When both versions of the gate template and both inputs are present, extension can occur. In all cases, T7 RNA polymerase can transcribe the aptamer on the extended template leading to a 1 signal. When neither input is present, the RNA polymerase is unable to transcribe. The lack of aptamer signal is recorded as a 0. **e** An OR gate truth table. **f** Fluorescent results of an OR gate encoding a Broccoli aptamer. On all panels, the value of each replicate within a sample set is represented by a gray marker. The green bars are the averages of each sample set (*n* = 3). All experiments were repeated three times independently. Source data are provided as a Source Data file.

18 to Base 105, extending from the latter half of the T7 Max promoter through to the entirety of the RNA aptamer. When both inputs are present, they will hybridize to each other, but parts of each strand will remain single-stranded. Notably, the T7 Max promoter sequence needs to be double-stranded in order for T7 RNA polymerase to bind the template and begin transcription. When both inputs hybridize in the presence of DNA polymerase (NEB OneTaq Polymerase), the enzyme extends each single-stranded portion of the input complex making the entire complex double-stranded. When the T7 Max promoter becomes double-stranded, the RNA polymerase can transcribe the RNA aptamer (Fig. 4a). The fluorescence of the RNA aptamer is recorded as a 1. When only one of the inputs is present with the DNA polymerase, extension cannot occur, and the T7 Max promoter sequence remains single-stranded. Transcription of the RNA aptamer cannot occur, and the lack of fluorescence is recorded as a 0 (Fig. 4b). The increase in fluorescence when both inputs are present is 42 times greater than when neither input is present (Fig. 4c).

The OR gate template is a total of 115 nucleotides and starts with 20 random bases, the T7 Max promoter, a sequence of the RNA aptamer, and ends with another 20 random bases (SI Fig. 11). In this case, the gate template is both the sense and antisense strands. Input 1 is a sense strand going from Base 1 to Base 39 and includes the first 20 random nucleotides and a part of the T7 Max promoter. Input 2 is an antisense strand going from Base 93 to Base 115 and includes the second set of random nucleotides and a small part of the RNA aptamer sequence. To validate the OR function where only one input is provided and the output is a 1, Input 1 is mixed with the antisense gate template or Input 2 is mixed with the sense gate template. In either case, the inputs will anneal to the complementary regions on gate

template strands. The provided DNA Polymerase will be able to extend the template from the input regions to create a double-stranded template. Most importantly, the T7 Max promoter sequence will become double-stranded, allowing T7 RNA polymerase to transcribe the template into the resulting RNA aptamer (Fig. 4d). The fluorescent signal of the aptamer is perceived as a 1 (Fig. 4e). When both inputs and both strands of the gate template are added together, twice as much gate template is polymerized into the double-stranded form, resulting in the increased concentration of transcribed RNA aptamer. This phenomenon is shown in Fig. 4f where the fluorescent signal when both inputs are provided is higher than the functions where only one input is provided. In contrast, when neither input is present, the sense and antisense gate template strands are processed in separate reactions to prevent self-hybridization. Because each strand remains entirely single-stranded, especially the T7 Max promoter, DNA polymerase does not extend the template and T7 RNA polymerase cannot transcribe the downstream RNA aptamer. The lack of fluorescence is recorded as a 1. The signal-to-noise ratio between the 1 and 0 signals is 9 to 1 for the OR gate.

## The circuits

The ultimate goal of the platform is to harness biological components and processes to create complex Boolean circuitry. After validating the function of each single gate, designing a multi-gate processor is crucial for demonstrating future potential. The NAND gate is widely known as a universal gate because it can be implemented in ways to create other Boolean operations without the use of other types of gates. Using three NAND gates in a specific pattern, we created an OR processor (Fig. 5a). NAND gate 1 (NAND 1) and NAND gate 2 (NAND 2) form the base of the

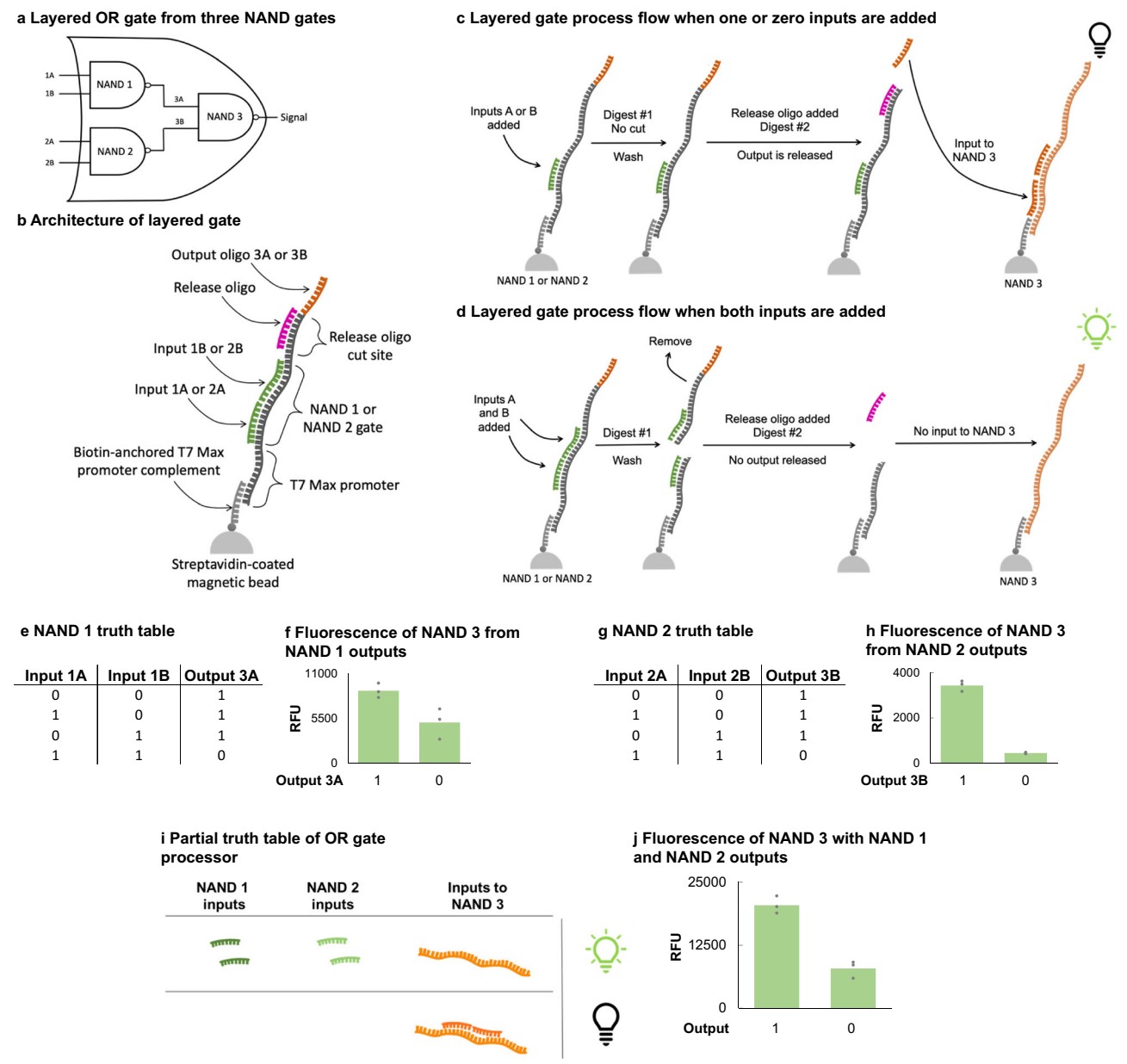

**Fig. 5 | A layered OR gate built from NAND gates. a** A schematic showing a full circuit of three NAND gates performing an OR gate operation. Within the processor, two unique NAND gates produce single-stranded DNA outputs, which become inputs for a third unique NAND gate. The final output of the third NAND gate is RNA aptamer fluorescence. **b** The general architecture of the first two NAND gates (NAND 1 or NAND 2) in the circuit. **c** When both pairs of inputs are present with NAND 1 or NAND 2, a restriction enzyme cleaves the gate templates, leaving the promoter-annealed portion of the templates attached to the magnetic bead. The cut portions of the templates can be removed by discarding the supernatant. When the release oligo is added to the digested versions of the gates, there is no complementary site for the release oligo to hybridize to. **d** When zero inputs are present with NAND 1 or NAND 2, the restriction enzyme does not cleave the gate template. The entire 106-base template remains attached to the beads, and remains bound to the magnetic beads. When the release oligo is added, it hybridizes with its complementary region on the gate template. When each output from NAND 1 and NAND 2 are added to NAND 3, they act as the single-stranded inputs and can hybridize with the NAND 3 gate template. The restriction enzyme can digest the template, which prevents the transcription of the downstream RNA aptamer. The lack of transcribed fluorescent signal is recorded as a 0. **e** NAND 1 truth table. (f) Fluorescence results of NAND 3 in all input conditions of NAND 1. (g)NAND 2 truth table. **h** Fluorescence results of NAND 3 in all input conditions of NAND 2. **i** Depiction of input conditions of NAND 1 and 2 where the outputs become the inputs for NAND 3. **j** Fluorescence results of the OR gate processor with all three NAND gates. On all panels, the value of each replicate within a sample set is represented by a gray marker. The green bars are the averages of each sample set (*n* = 3). All experiments were repeated three times independently. Source data are provided as a Source Data file.

OR gate. They are each composed of unique gate templates and inputs following an architecture similar to those of the single NAND gates mentioned in Fig. 1. However, instead of a fluorescent RNA aptamer output, NAND 1 and NAND 2 each output 15-base single-stranded DNA sequences. These two output sequences will become the inputs for NAND gate 3 (NAND 3). The combined functions of NAND 1, 2, and 3 form an OR gate.

The required T7 Max sense strand, which is complementary to the promoter region on the NAND gate templates, is conjugated to biotin. When the biotinylated promoter complement is bound to magnetic beads coated with streptavidin, any sequence hybridized with the promoter complement (i.e., the promoter sequence on the gate template) will also be bound. As the magnetic beads are immobilized, the DNA strands are also consequently immobilized (Fig. 5b). The

immobilization aspect is a tenet of processor function on this platform. Immobilization of the NAND gates mimics the 2D nature of more traditional computing circuitry, while simultaneously enabling the exchange of digested gate templates, complementary oligos, and output oligos. The gate templates for NAND 1 and NAND 2 start with a biotin molecule followed by a spacer region of 7 random bases, the antisense T7 Max promoter sequence, an antisense gate region of 10 random bases, a restriction enzyme recognition site, and another 10 random bases, an antisense release oligo region of 6 random bases and the same restriction enzyme recognition site, and an output oligo region. The output oligo regions are each of the sense inputs for NAND 3 (SI Figs. 12 and 13). To accommodate the antisense directionality of the NAND 1 and NAND 2 gate templates, this sense input is reversed in direction before it is added to the gate templates. When the outputs are released, they will be able to hybridize with the NAND 3 gate template in the correct direction (5′ − 3′). The NAND 3 gate template contains an antisense T7 Max promoter sequence, an antisense gate region that is complementary to the output oligos released from NAND 1 and NAND 2, and the antisense sequence for an RNA aptamer (SI Figure 14). The final fluorescent output of NAND 3 determines the outcome of the entire OR processor−a higher fluorescent signal signifies a 1 and a lower fluorescence signifies a 0.

NAND 1 and NAND 2 function in the same way, despite containing unique sequences and releasing unique output oligos. Inputs for each gate were designed very similarly to those designed for the single NAND gates. After the NAND 1 and 2 gate templates have annealed to the streptavidin bound, biotinylated T7 Max promoter complement, inputs can be added along with the chosen restriction enzyme. Pairs of inputs for each gate are added depending on the outcome desired (e.g., NAND 1 gate inputs, 1A and 1B, are added together or not at all). Inputs have the potential to be added singularly to either NAND 1 or NAND 2 in order fulfill all aspects of a truth table, but due to the complexity of the processor, we simplified experimental sample types. NAND 1 and NAND 2 gate reactions were spatially separated into different reaction vessels for the initial studies reported here.

When zero pairs of inputs are added, the restriction enzyme cannot digest either of the templates, leaving the entirety of both sequences still immobilized to magnetic beads (Fig. 5c). A magnetic plate is used to separate bead-bound sequences from those floating freely in the supernatant. In this case, the supernatant only contains the restriction enzymes which are removed through a wash step. Another solution containing the 12-base sense complement to the release oligo regions on the gate templates is added to the immobilized bead fractions. Because the gate templates were not digested and removed, the release oligo complement can hybridize to the correct areas on the templates−the areas immediately downstream of the gate regions. When the restriction enzyme corresponding to the cut site encoded in the release oligo is added to the NAND 1 and 2 reactions, the restriction enzyme cleaves the templates. The 15-base output oligos are released from the gate templates into the supernatants. Both supernatants are added to the NAND 3 gate reaction, where each output oligo in the supernatants from NAND 1 and NAND 2 act as inputs for the final gate (Fig. 5e−h). NAND 3 operates like previously mentioned single gates, in that when both inputs are provided, the gate template is digested, and no fluorescent RNA aptamer is transcribed. The lack of fluorescence signifies a 0 signal output for the OR gate processor where zero starting inputs (i.e., inputs 1A, 1B, 2A, and 2B) were added (Fig. 5i).

When both pairs of inputs are added to each immobilized gate template, the restriction enzyme digests both templates (Fig. 5d). The supernatants of each gate reaction will now contain the latter half of the gate templates, where the release oligo and output oligo regions are. When the supernatants are removed and discarded, only the truncated gate templates still annealed to the biotinylated-T7 Max sense sequence remain immobilized to the beads. When the solution containing the release oligo complement is added to the beads, there are no sequences for the release oligo to anneal to. There are no double-stranded release oligo cut sites for the next restriction enzyme to digest, so no output oligos are released into the supernatants for addition to the NAND 3 gate reaction (Fig. 5e−h). Although the supernatants from NAND 1 and NAND 2 are still added to NAND 3, the lack of output oligos (i.e., inputs for NAND 3) from the previous gates prevent the gate region from becoming double-stranded. The restriction enzyme cannot digest NAND 3 and T7 RNA polymerase will transcribe the entire template including the RNA aptamer encoded in the template. High fluorescent signal of the RNA aptamer signifies a 1 output for the entire OR processor, where both pairs of inputs were added (Fig. 5i).

Figure 5 shows data for NAND 1, NAND 2, and NAND 3 gate templates designed with PvuII restriction enzyme recognition sites. PvuII was used for recognition sites in the gate regions and the release oligo regions. There is a 2.6 times increase in signal when neither input is added to NAND 3 from the NAND 1 and NAND 2 reactions (a 1 signal), compared to when both inputs are added to NAND 3 (a 0 signal) (Fig. 5j). There is still further optimization required to improve the signal to noise ratio of this multi-gate processor. In our preliminary efforts to optimize the signal differences between 0 and 1, we discovered that higher starting reagent concentrations were required for the NAND 1 and NAND 2 reactions. The excess in starting reagents ensures that the concentrations of final inputs for NAND 3 are higher than the concentration of the gate template. This allows the final gate to function properly. We don't see this as a major detriment to the platform because similar system requirements exist for primitive electrical circuits as well, where input voltages may have to be increased to generate acceptable output voltages[35]. Since biocomputing is following the lead of electrical engineering and computer science in many ways, this particular characteristic of a multi-gate Trumpet processor is reminiscent of quirks in early versions of silicon-based computing.

## Discussion

In this work, we demonstrated a new principle for biocomputing: a platform combining advantages of in vitro and live cell logic operations. We have validated Trumpet performance with four different types of readout, several enzymes, and dozens of logic gate sequences. The web-based Trumpet script enables streamlined design of Trumpet logic gate sequences.

The current capacity of Trumpet is, using traditional computing analogies, closer to the Ishango bone than a microprocessor. We have demonstrated the capacity of Trumpet to perform all basic types of Boolean logic gates, and the rudimentary capacity for layering the gates into a larger processor. More work is needed to solve remaining technical challenges on the way to wider utilization of Trumpet as a biocomputing operating system. Namely, template regeneration and technical methods for scaling down reaction volumes will be the next two largest milestones.

The Trumpet "OS" is not self-replicating like cell-based logic gate systems and is more sensitive to temperature and reaction conditions than simpler technologies based on small molecules and nucleic acids. However, Trumpet is more programmable and predictable than live cells, with better signal amplification and reaction fidelity than simple non-enzymatic methods.

Biological computing is growing in significance, particularly as new needs and applications arise in data storage, self-regenerative biomedical devices, biomanufacturing, and autonomous devices. The advancements in bio-inspired design accelerated many applications for biological processors. While no biocomputing technology matches the speed, reliability, and scalability of traditional operating systems at this present moment, the effort devoted to developing this field and the many needs for this technology will drive fast progress. As in the

case of traditional computing, the next generation of biocomputing technologies will appear on several different platforms. Trumpet is designed to fill the need and application gap between simplest biochemical logic gates and the more autonomous live cell-based circuits. Together, all these technologies offer a comprehensive toolbox of biocomputing solutions for a variety of cutting-edge applications.

## Methods

### Fluorescence data
Fluorescence data was collected using plate reader software SoftMax Pro v5.4. The data were analyzed using MS Office Excel 2016 and Igor Pro 8.

### Designing a Boolean logic gate
Benchling was used initially as the design platform for manually building a logic gate. The first gate, a NAND gate, was designed by concatenating the promoter sequence for T7 Max RNA Polymerase (5′-AATTCTAATACGACTCACTATAGGGA-3′) with 10 randomly generated DNA bases, a 6-base restriction enzyme's recognition sequence, another 10 randomly generated DNA bases, and an RNA aptamer's antisense DNA sequence. The gate template is the entire antisense strand, and the inputs are each 13 bases of the sense strand. Each input spans from one set of the randomly generated bases to half of the recognition site (SI Fig. 3). To randomly generate the 20 nucleotides flanking the restriction enzyme's cut site, the Random DNA Sequence Generator (http://www.faculty.ucr.edu/~mmaduro/random.htm) was used with 50% GC content settings. The restriction enzymes employed on the logic gates were chosen according to their continued functionality in a newly developed logic gate buffer, aHOT 7.9, and their ability to withstand temperatures over 90 °C. The NEBuffer Activity/ Performance Chart with Restriction Enzymes was used for choosing restriction enzyme candidates (https://www.neb.com/tools-and-resources/usage-guidelines/nebuffer-performance-chart-with-restriction-enzymes). In most of the logic gates discussed in this paper, the restriction enzyme *PvuII* was used, and its recognition cut site (5′-CAGCTG-3′) was built in between the randomly generated flanking sequences. aHOT 7.9, the logic gate buffer, was modeled after New England BioLabs Inc.'s One*Taq* 2X Master Mix with Standard Buffer (Catalog No. M0482) and contains several additional reagents to support restriction enzyme digestions, polymerase reactions, and cell-free transcriptions. The 5X aHOT 7.9 buffer contains 100 mM Tris-HCl, 120 mM $MgCl_2$ hexahydrate, 110 mM $NH_4Cl$, 500 mM KCl, 0.3% IGEPAL CA-630, 0.25% Tween-20, 5 mM Spermidine, and 5 mM dithiothreitol (DTT), adjusted with HCl to pH 7.9. The detergents IGEPAL CA-630 and Tween-20 were added after pH 7.9 was achieved.

A NOT gate was designed by concatenating a T7 Max promoter sequence with 10 randomly generated bases, a 6-base restriction enzyme recognition site, another 10 randomly generated bases, and an RNA aptamer sequence. The gate template is the entire antisense sequence, while the input is the 26-base sense sequence from the first set of random bases through to the second set of random bases (SI Fig. 8).

The design for a NOR gate contains the T7 Max promoter sequence, two 26-base gate regions (each with restriction enzyme recognition sites flanked by 10 randomly generated nucleotides), and the RNA aptamer sequence. The gate template is the antisense sequence and each of the two inputs are the sense sequences of each of the 26-base gate regions (SI Fig. 9).

The AND gate contains the T7 Max promoter, 30 random bases, and the RNA aptamer sequence. One input is part of the sense strand, and the other input is part of the antisense strand (SI Fig. 10). The length of each input was chosen so that the annealing temperature was between 65 and 68 °C when using OneTaq Polymerase in Standard Buffer and default concentrations.

The OR gate contains the T7 Max promoter, followed by the RNA aptamer sequence. The gate template is both the sense and antisense strands, while each input is either the sense or antisense strand (SI Fig. 11). The input lengths for this gate were chosen so that the annealing temperatures were 57 °C.

All designed gate sequences, inputs, and complementary sequences can be found in Supplementary Data file 3. All oligomers related to a gate reaction were ordered through Integrated DNA Technologies (IDT). Some gate templates that exceeded 100 nucleotides in length were ordered as 4nmol Ultramers. All oligomers were ordered in lyophilized conditions and rehydrated to concentrations of 100 μM prior to experimental use. All oligomers were used without additional purification unless otherwise stated.

### Trumpet browser platform
Trumpet is a website designed to generate DNA sequences encoding Boolean logic (i.e., AND, OR, NAND, NOT, and NOR) gates. Our Github repository can be found at https://github.umn.edu/kadamala/ Trumpet. Upon entering https://trumpet.bio, a request is sent to our NGINX server which acts as a reverse proxy to serve our WordPress site. The WordPress site contains all the content for Trumpet including the Build page where you can construct your own transcriptional Boolean logic gates. On the build page, you can choose which promoter, restriction enzyme, and output modality the Trumpet Sequence will use, the GC Percentage and melting temperature the Trumpet Sequence will have, and the number of Trumpet Sequences you wish to generate. Pressing the "Build" button will then send an HTTP request to our Flask server with the previously mentioned sequence configurations. The Flask server then constructs a potential sequence using the selected promoter, restriction enzyme cut site, output option, and randomly generated strands flanking the restriction enzyme cut site. We use Python 3's Random library, which provides a pseudo-random number generator, for all randomness used in Trumpet (https://docs.python.org/3/library/random.html#random. choice). The randomly generated strands and the restriction enzyme cut site encode a chosen logic gate. Trumpet then runs the potential sequence through a local version of mFold 3.6 (http://www.unafold. org/mfold/software/download-mfold.php) and NUPACK 4.0 (http:// www.nupack.org/downloads). NUPACK provides a DPP notation of the RNA fold (https://docs.nupack.org/definitions/#dot-parens-plus-notation). With mFold, we generate the DPP notation from the provided base pairings. For example, mFold might report that base 1 is paired with base 5, base 2 is paired with base 4, and base 3 is unpaired which results in ((.)) as DPP notation. We then see if the reporter of the sequence folded properly by checking if the DPP notation contains a target structure, especially if a fluorescent RNA aptamer output is requested. The secondary structure of a transcribed gate template in DPP notation could look like........(((...(((((.(........).)))))))) (((((((.((((((((.......)))))...)).)))))))). The bolded portion is the target structure we are looking for, which indicates that the Broccoli, the RNA aptamer in this example, is folding correctly. If the output folds properly, the sequence is valid. Trumpet continues to generate potential sequences and run them through mFold and NUPACK until enough valid sequences have been found. An HTTP response containing all the valid sequences is sent back to the WordPress site where a CSV file is automatically downloaded for the user. The CSV file contains the antisense strands of the Trumpet sequences (referred to as "gate templates" in the Methods) and the sense strands of the inputs which are complementary to the randomly generated flanking sequences and the restriction enzyme cut sites in the Trumpet sequences.

### Logic gate reactions
For all gates, there are two sets of reactions that need to take place: Reaction A (restriction enzyme digestions or polymerase reactions)

followed by Reaction B (cell-free transcription with fluorescent readout).

A typical Reaction A of a restriction enzyme digestion for a single NAND gate looks like 1 µL of a restriction enzyme, 3 µL of 5X aHOT 7.9 buffer, 1 µL of 25 µM gate template, 1 µL of 25 µM T7 Max promoter sense complement, 3 µL of 25 µM input 1 and 3 µL of 25 µM input 2 when applicable, and ddH$_2$O to bring the total volume up to 15 µl. The water and reagent volumes of regular restriction enzyme digests were omitted or reduced, respectively, to compensate for templates volumes that would be necessary for the cell-free transcription in Reaction B. The template for Reaction B is the entire volume of Reaction A to avoid overly diluting the transcription reagents by adding the usual 25 µL restriction digests or waste transcription reagents by increasing their concentrations proportionally. A single NAND gate Reaction A was subjected to a short annealing program (95–37 °C in 5 °C per minute increments) and then incubated in a thermocycler (Bio-Rad C1000 Touch) at 37 °C for 15 min.

A typical NOT gate reaction consists of 1 µL of a restriction enzyme, 3 µL of 5X aHOT 7.9 buffer, 1 µL of 25 µM gate template, 1 µL of 25 µM T7 Max promoter sense complement, 3 µL of 25 µM input when applicable, and ddH$_2$O to bring the volume up to 15 µl. Reaction A was subjected to the same annealing and digest incubation protocols mentioned above.

A NOR gate Reaction A is very similar to the NOT gate except for the addition of the second input, which will reduce the amount of ddH$_2$O required in the final reaction volume. The same reaction protocols were followed as for the NAND gate.

The AND gate Reaction A utilizes New England Biolabs OneTaq Polymerase (Catalog No. M0480) PCR recommendations. However, instead of running the reaction for 30 cycles, the AND gate only requires one cycle. Each AND gate reaction has a final volume of 25µL and includes 5µL of OneTaq 5X Standard Reaction Buffer (NEB Catalog No. B9022S), 1 µL of 100 µM Input 1, 1 µL of 100 µM Input 2, 0.5 µL of 10 mM dNTPs (NEB Catalog No. N0447S), 1 µL of OneTaq Polymerase, and 17µL of ddH$_2$O. Because each input was designed so that the annealing temperature was 66 °C, the annealing and extension temperatures and times were combined to have a single incubation at 68 °C for 30 min (Bio-Rad C1000 Touch). The incubation time is much longer than probably necessary for OneTaq Polymerase, but there is at minimum 4 µM of DNA in each reaction, so a lengthy incubation time seemed somewhat appropriate.

The OR gate Reaction A also uses OneTaq Polymerase with similar PCR reaction master mix reagents. For a final reaction volume of 25 µM, a single OR Reaction A includes 5 µL of OneTaq 5X Standard Reaction Buffer (New England BioLabs Catalog No. B9022S), 1 µL of 100 µM sense gate template, 1 µL of 100 µM antisense gate template, 1 µL of 100 µM Input 1, 1 µL of 100 µM Input 2, 0.5 µL of 10 mM dNTPs (NEB Catalog No. N0447S), 1µL of OneTaq Polymerase (NEB Catalog No. M0480), and 14.5 µL of ddH$_2$O if all of the reagents were added in these concentrations. For the reactions where only one of the gate template types (sense or antisense) and input were used, the loss in volume was compensated by adding an equivalent value of ddH$_2$O. Since the annealing temperature of the OR gate inputs was much lower than that of the AND gate inputs, two different incubation temperatures is required. Incubation started at the annealing temperature of 57 °C for 5 min and then proceeded to the extension temperature of 68 °C for 30 min (Bio-Rad C1000 Touch).

## Cell-free transcriptional signal readout

All gate Reaction As were used as the template for a cell-free transcription reaction that produces a fluorescent RNA aptamer. This transcription reaction will be referred to as Reaction B in this Methods. Typical cell-free transcription reactions are quite compact in volume (<20 µL), but because Reaction A volumes were a minimum of 7 µL and often contained highly concentrated gate templates, the concentration of reagents were increased to compensate. As such, for a final cell-free transcription reaction volume of 54 µL, each reaction contained 12 µL of aHOT 7.9 5X Buffer, 12 µL of 20 mM NTPs (Larova GmbH Ribonucleotides), 6 µL of 1 mM DFHBI (if Broccoli is the intended RNA Aptamer, otherwise 100 µM of any other ligand), 7 µL of Reaction A template, 7.5 µL of ddH$_2$O, 6 µL of 1.5 µM T7 RNA Polymerase, 6 µL of Inorganic Pyrophosphatase (Bayou Biolabs Catalog No. E-108), and 0.5 µl of RNase Inhibitor (NEB Catalog No. M0314).

T7 RNA polymerase was overexpressed and purified internally in our laboratory. 10 mL LB containing 100 µg/µl carbenicillin was inoculated with *E. coli* DH5α containing pT7-911Q (T7 RNAP)[36]. The culture was grown overnight at 37 °C, then used to inoculate an additional 1 L of LB containing 100 µg/µl carbenicillin and grown at 37 °C to an OD600 between 0.5 and 1. The culture was then induced with 1 mM IPTG and grown at 37 °C for 3 h. The culture was cooled on ice for 20 min and pelleted at 2650 × g for 15 min. The pellet was flash-frozen in liquid nitrogen and frozen at −80 °C overnight. The pellet was held in a cold room for 30 min, then dissolved in 20 mL lysis buffer (50 mM HEPES-KOH pH 7.6, 1 M NH$_4$Cl, 10 mM MgCl2, 7 mM BME). The pellet was incubated in lysis buffer for 30 mins followed by tip sonication. Sonication was performed at 50% power in 15 s intervals until 2 kJ total energy had been applied, then the sample was allowed to cool for 5 min. This was repeated a total of 4 times. The pellet was then centrifuged for 45 min at 15,000 × g at 4 °C. The supernatant was applied to 0.6 mL Ni-NTA agarose beads (GoldBio, H-350-50) and incubated on a rocker in a cold room for 1 h. Washing and elution steps were done in batch method. Beads were washed with 10 mL wash buffer for 10 mins then washed again with 10 mL wash buffer (50 mM HEPES pH 7.6, 1 M NH$_4$Cl, 10 mM MgCl$_2$, 15 mM imidazole, 7 mM BME) for 15 min. 3 mL elution buffer (50 mM HEPES-KOH pH 7.6, 100 mM KCl, 10 mM MgCl$_2$, 300 mM imidazole, 7 mM BME) was applied to beads and incubated on a rocker for 12 min in a 4 °C cold room. Elution was dialyzed against 500 mL 2X storage buffer (100 mM Tris-HCl pH 7.6, 200 mM KCl, 20 mM MgCl$_2$, 14 mM BME) using Slide-Alyzer Dialysis Cassette, 2000 MWCO (Thermo Fisher Scientific, 66203) overnight, followed by dialysis against an additional 500 mL 2 X storage buffer for 3 h. Because this enzyme was intended for lyophilization, it was prepared in the same storage buffer with the omission of glycerol. T7 RNA Polymerase was quantified using the calculated A280 on a NanoDrop ND-1000. Protein activity was assessed by in vitro transcription of Broccoli aptamer and kinetic monitoring on a fluorescence plate reader (T7 RNAP).

For Broccoli transcription, DFHBI (4-[(3,5-difluoro-4-hydroxyphenyl)methylidene]−1,2-dimethyl-4,5-dihydro-1H-imidazol-5-one, Tocris Catalog No. 5610) is the appropriate ligand. Broccoli can also bind DFHBI-1T (https://www.tocris.com/products/dfhbi-1t_5610). The Pepper aptamer can bind a suite of HBC ligands, which fluoresce at varied wavelengths. In this study, we used Pepper with the ligand HBC-620 (4-((2-hydroxyrthyl)(methyl) amino)-benzylidene)-cyanophenyl-acetonitrile). For Corn, DFHO (3,5-difluoro-4-hydroxybenzylidene imidazolinone-2-oxime, Tocris Catalog No. 6434) is the appropriate ligand. For Mango, TO1-PEG-biotin (ABM Catalog No. G955) is the appropriate ligand. For Malachite Green, Malachite Green Dye (Sigma Aldrich Catalog No. M9015) is appropriate.

After reaction preparation, all cell-free reactions were aliquoted into 384-well black, clear-bottom spectrophotometer plates (Sigma Aldrich Catalog No. M6811). Molecular Devices Gemini EM spectrophotometers were used for both incubation at 37 °C and bottom-read fluorescence capture of all RNA aptamers mentioned in this study. Cell-free transcription reactions were incubated at 37 °C for 5 h minimum with excitation and emission for fluorescence capture occurring every 30 min. The excitation/emission wavelengths for each RNA aptamer used in this study is as follows: Broccoli[26] 472 nm/507 nm, Pepper with HBC620[28] as the ligand 580 nm/620 nm, Mango[29] 510 nm/535 nm, Corn[30] 505 nm/545 nm, and Malachite Green[31] 630 nm/650 nm.

Standard error of the mean (SEM) was calculated for all experiments in this study. SEM was calculated as

Standard deviation $\sigma = \sqrt{\frac{\sum_{i=1}^{n}(x_i - \bar{x})^2}{n - 1}}$ where $\bar{x}$ = sample mean and n = sample size

Standard error $(\sigma_{\bar{x}}) = \frac{\sigma}{\sqrt{n}}$

The individual values of each triplicate are reported as gray markers on each bar graph in the main text of the manuscript.

## Multi-gate OR processor

The OR gate processor experiments occurred in three stages: Stage 1 is where inputs were added to NAND 1 and NAND 2, Stage 2 is where the release oligo was added to all gate reactions, and Stage 3 is where supernatants from NAND 1 and NAND 2 reactions were added to the final, NAND 3 reaction. The output of NAND 3 is RNA aptamer fluorescence.

Two different gates for both NAND 1 and NAND 2 in Stage 1 of the experiment were multiplexed and set up as follows. 35 µL of room temperature 100 µM biotinylated T7 Max complement was adhered to 35 µL of room temperature magnetic beads coated with streptavidin (NEB Catalog No. S1420S). After incubation at room temperature for 15 min, the beads and the attached T7 Max complementary sequences were immobilized on a 96-well magnetic plate (Alpaqua SKU A001322). The supernatant was removed and discarded, and the sequences were removed from the magnetic plate. The pelleted T7 Max complement was resuspended in 35 µl of ddH$_2$O for an assumed concentration of 100 µM and was used for most of the multi-gate experiments. A NAND 1 reaction contained 1 µl PvuII restriction enzyme, 3 µl NEB r3.1 Buffer (10X) (NEB Catalog No. B6003S), 2.5 µl of each 100 µM gate template, 5 µl of 100 µM biotinylated-magnetized T7 Max complement sequence, 2.5 µl of each 100 µM Input 1 A, Input 1B, Input 2A, and Input 2B for a final volume of 24 µl. For the samples not containing any inputs, we compensated for the loss in volume with equivalent volume of ddH$_2$O (10 µl in this case). All Stage 1 reactions were incubated in a thermocycler (Bio-Rad C1000 Touch) at 37 °C for 20 min with a cool down to 12 °C.

All oligomers in the NAND 1 and NAND 2 reactions were at a concentration of 10 µM. These concentrations were the current protocol at the time of submission of this study. However, further optimization may change the gate template and input concentrations reported here.

For Stage 2, all multiplexed NAND 1 and NAND 2 reactions were removed from the thermocycler. After placing the samples into the magnetic plate, the supernatant was removed from the samples that contained inputs. The pellets in these samples were resuspended with 24 µl of ddH$_2$O. The supernatants of samples without inputs was not removed. A Stage 2 master mix was prepared containing 1 µl of PvuII restriction enzyme, 1 µl NEB r3.1 Buffer (10X), and 3 µl of 100 µM release oligo for each sample. 5 µl of the Stage 2 master mix was added to each reaction and mixed thoroughly. Stage 2 reactions were incubated (Bio-Rad C1000 Touch) for another 20 min at 37 °C with a cool down to 12 °C. The samples were then transferred to a SpeedVac vacuum concentrator (Thermo Scientific Savant SPD1010) to concentrate the multiplexed samples overnight.

In Stage 3, all Stage 2 reactions were removed from the vacuum concentrator and resuspended with 5 µl of ddH$_2$O and then placed in the magnetic plate. Prior to removal of the supernatant, NAND 3 reactions were prepared containing 1 µl of PvuII restriction enzyme, 1.5 µl of aHOT 7.9 Buffer (10X), 2 µl of 100 µM NAND 3 gate template, and 2 µl of either 100 µM biotinylated T7 Max complement or non-biotinylated T7 Max complement. For the NAND 3 reactions not intended to have any inputs, add the 5 µl supernatants from the Stage 2 concentrated resuspension that do not contain the outputs 3A or 3B. For the reactions that are supposed to have inputs, add the supernatants that do contain outputs 3A and 3B. The final concentration of inputs in this Stage 3 reaction, assuming all restriction enzyme digests

were completely efficient, is 15 µM. The NAND 3 gate template in the Stage 3 reactions were 12 µM. All Stage 3 reactions were incubated in a thermocycler for 20 min at 37 °C with a cool down to 12 °C.

After incubation, all Stage 3 reactions were used as templates in cell-free transcription reactions. Transcription reaction protocol was followed as written in section "Cell-free Transcriptional Signal Readout". Fluorescence was measured as kinetic readings over 5 h at 37 °C in 30 min increments. SEM was calculated as written previously for each set of triplicates. However, the individual values of each triplicate are reported as gray markers on each bar graph in the main text of the manuscript.

## Reporting summary

Further information on research design is available in the Nature Portfolio Reporting Summary linked to this article.

## Data availability

All data is available in supplementary information files accompanying this manuscript. Source data are provided with this paper.

## Code availability

The code of the Trumpet website is available in the Supplementary Software file.

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

## Acknowledgements

We sincerely thank John Berchmans and David Gonzalez for thoughtful early discussions about computing and comments on this manuscript. This work was supported by NSF award 1807461 (to K.P.A.), NSF award 1844313 (to K.P.A.), generous gift from Jeremy Wertheimer (to K.P.A.), Hackett Royalty Fund award (to K.P.A.), Alfred P. Sloan Foundation grant G-2022-19420 (to K.P.A.), the ARCS Foundation Scholarship (to J.S.), Scialog grant #28754 from Research Corporation for Science Advancement (to A.E.E.), grant #2021-3123 from the Heising-Simons Foundation (to A.E.E.), and NASA Contract 80NSSC18K1139 under the Center for Origin of Life (to A.E.E. and K.P.A.).

## Author contributions

K.P.A., A.E.E., and J.A.S. designed the study, performed experiments, analyzed data, and wrote the manuscript. C.D. and M.B. performed experiments and analyzed data. A.D. and A.S. contributed equally and built most of the functions in the Trumpet browser platform. S.O. developed the backend of the Trumpet browser platform. L.M.A. and A.E.E. assisted in early designs of the logic gates.

## Competing interests

The authors declare the following competing interests: authors who conceived this study and performed bulk of the experiments, J.A.S., A.E.E. and K.P.A., filed a patent application with the United States Patent and Trademark Office, application number: 18124864. C.D., A.F., A.S., M.B, S.O.B. and L.M.A. declare no competing interests.
