## [Peer Review File · Nature Communications]

Trumpet is an operating system for simple and robust cell-free biocomputingReviewers' Comments:

Reviewer #1:

Remarks to the Author:

The article by Sharon et al. describes a new biomolecular computing system based on in-vitro components. Specifically, Trumpet makes use of restriction enzymes and polymerases to construct a broad range of Boolean logic gates. The system is simple in design and functions robustly. In addition, by providing a web-based tool, the system can be simply used by other researchers in the field making it a broad tool that can be used for example in molecular diagnostics. The article can be improved in several ways:

- 1) While biomolecular computing systems have many applications, specifically in diagnostics, I do not see them as direct competitors for electronic circuits as the abstract and introduction suggest. The speed of these type of circuits is just too slow. As such, I would suggest the authors remove any claims where biomolecular computing systems are suggested as an alternative to electronic circuits.
- 2) In most Figures bar graphs are used to present results of multiple measurements. I would suggest the authors plot the data points for all cases instead of only the mean and SEM. This will give a clearer insight into the spread of the different data points.
- 3) A better description should be given on how replicates were performed. Did all replicates use the same batch of enzymes and stock solutions? If all replicates have been obtained by using the same batch of enzymes and/or stock solution, I would like to see the effect of batch-to-batch variation in the enzymes and preparation of different stock solutions.

After these comments are considered the article is of sufficient quality to be accepted.

Reviewer #2:

Remarks to the Author:

In this work, the authors demonstrated an enzymatic system that can process signals of DNA inputs in Boolean logic gates. They implemented NAND, NOT and NOR gates using restriction enzymes, together with AND and OR gates using DNA polymerases. Moreover, they also developed a web-based platform for gate sequence design and built an OR processor with three NAND gates. This enzymatic induced logic gating points out to a new approach for biocomputing.

Specific comments:

1. Major optimizations are expected to improve the performances of the logic gates (output signals), either by gate sequence design or gate architecture design. For instance, strong crosstalk between Template A and Pair F in Figure 1m, only 3x signal difference in the NOT gate (Figure 3C), and only 1.4x signal change in the layered OR gate (Figure 5j).
2. The reactions were demonstrated mostly in micromolar concentration, which would be a limiting factor for potential future applications in data storage, biomedical and autonomous devices. Is it possible to push for a higher sensitivity?
3. The demonstration of the layered OR gate with three NAND gates (two of them are same) is rather simplified. Is it possible to build more complex Boolean circuitry using this method?
4. What are the advantages of this enzymatic method compared to the strand displacement method? Since multiple enzymes are involved in this system, there would be gaps between this method and strand displacement method when considering the reaction speed and the cost.
5. There is a typo in Figure 1b, the last two rows in the NAND truth table are the same.

We sincerely thank the Reviewers for all the constructive comments and suggestions that improved this article in all conceptual and experimental aspects.

We addressed all questions and comments with new experiments, changes in the manuscript and with point to point responses below.

Our responses to Reviewer comments are marked in green.

Reviewer #1 (Remarks to the Author):

The article by Sharon et al. describes a new biomolecular computing system based on in-vitro components. Specifically, Trumpet makes use of restriction enzymes and polymerases to construct a broad range of Boolean logic gates. The system is simple in design and functions robustly. In addition, by providing a web-based tool, the system can be simply used by other researchers in the field making it a broad tool that can be used for example in molecular diagnostics. The article can be improved in several ways:

We thank the Reviewer for all the feedback and comments that helped to improve this manuscript.

1) While biomolecular computing systems have many applications, specifically in diagnostics, I do not see them as direct competitors for electronic circuits as the abstract and introduction suggest. The speed of these type of circuits is just too slow. As such, I would suggest the authors remove any claims where biomolecular computing systems are suggested as an alternative to electronic circuits.

We thank the Reviewer for pointing this out, we reconsidered this issue. We have removed the claim of electronic circuits alternative from the manuscript.

2) In most Figures bar graphs are used to present results of multiple measurements. I would suggest the authors plot the data points for all cases instead of only the mean and SEM. This will give a clearer insight into the spread of the different data point.

We have corrected that. All bar graphs now show individual data points transposed onto each bar value, which is the average of that sample set.

3) A better description should be given on how replicates were performed. Did all replicates use the same batch of enzymes and stock solutions? If all replicates have been obtained by using the same batch of enzymes and/or stock solution, I would like to see the effect of batch-to-batch variation in the enzymes and preparation of different stock solutions.

Since these experiments happened over a 3 year period, we used several different batches of restriction enzymes, T7, RNase inhibitors, and inorganic pyrophosphatase. The aHOT 7.9 buffer was also made in the lab at least 3 times (3 batches). Each batch was 15ml, and was aliquoted

into 1ml tubes for storage in -20C. A batch lasted from 6 - 8 months depending on the frequency of experimentation. We have not observed significant batch to batch variability.

Unlike in vitro translation (where different batches of extract can produce drastically different results), it appears that the restriction enzyme activity and transcription used in this work are not as susceptible to variability based on different buffer stocks and batches of enzyme.

After these comments are considered the article is of sufficient quality to be accepted.

Reviewer #2 (Remarks to the Author):

In this work, the authors demonstrated an enzymatic system that can process signals of DNA inputs in Boolean logic gates. They implemented NAND, NOT and NOR gates using restriction enzymes, together with AND and OR gates using DNA polymerases. Moreover, they also developed a web-based platform for gate sequence design and built an OR processor with three NAND gates. This enzymatic induced logic gating points out to a new approach for biocomputing.

We thank the Reviewer for the feedback and comments that helped improve this manuscript.

Specific comments:

1. Major optimizations are expected to improve the performances of the logic gates (output signals), either by gate sequence design or gate architecture design. For instance, strong crosstalk between Template A and Pair F in Figure 1m, only 3x signal difference in the NOT gate (Figure 3C), and only 1.4x signal change in the layered OR gate (Figure 5j).

Figure 1m has been edited. After looking into the raw data for Template A and Pair f, we discovered that there was a signal difference between the template's true, matched inputs and the mismatched Pair F. However, the values were not comparable to the signal values found between other templates and mismatched inputs. Because Template A and Pair F could be suffering from nucleotide-level kinetic issues that we cannot delve at this time, we redesigned the gate templates and inputs. The supplementary document containing all the heatmap data (SI9) was also edited with this additional data.

Figure 3c was edited to contain new NOT gate data.

To address the signal difference issues of Figure 5j, we tested two new designs of the NAND 1 and NAND 2 gates. While we tested out the new and old designs in a few other troubleshooting attempts, we finally tried either multiplexing both NAND 1 and NAND 2 gates or concentrating double volumes of the starting gates through lyophilization. While those techniques separately did not work, we discovered that multiplexing two different designs of the NAND 1 and NAND 2 gates and vacuum concentrating the reactions into smaller volumes led to the desired outcome of increasing the signal to noise ratio. In the manuscript, we now report a signal to noise ratio of 2.6 (where it was previously 1.4). The Methods section has also been edited with the new protocol for multiplexing and concentration.

2. The reactions were demonstrated mostly in micromolar concentration, which would be a limiting factor for potential future applications in data storage, biomedical and autonomous devices. Is it possible to push for a higher sensitivity?

We provided data from an experiment where nanomolar concentrations were used for a NAND gate. That experiment showed that we can get down to 100nM of gate template while still seeing differences between the 0 and 1 signal. However, the global levels of fluorescence decrease as the concentration of the gates and inputs decrease. This effect is also shown in entirety in SI Figure 8.

We did an additional experiment to show whether it's possible to use gate templates at 10nM concentrations. We added experimental data to SI Figure 8.

3. The demonstration of the layered OR gate with three NAND gates (two of them are same) is rather simplified. Is it possible to build more complex Boolean circuitry using this method?

All three NAND gates in the layered circuit are unique in gate and input sequence. Annotated sequences are shown in SI Figures 14, 15, and 16. Theoretically, it is possible to build even more complex circuitry, especially if we involve NOT and NOR gates since they already use restriction enzymes for gate operation. The way that each of these gates are designed lends to the idea that we can have a DNA output that becomes the input for the next subsequent gate.

The output functionality of the AND and OR could also be redesigned to include a DNA output using a restriction enzyme after the DNA polymerases have acted on the gates. This type of redesign could work if we wanted to use AND and OR gates in the specific layered gate process we developed (i.e. biotinylated T7 Max promoter sequence, wash steps, etc.).

We are showing a relatively simple layered gate as a proof of concept. Future work expanding this system will investigate the limits of possible complexity of layering of multiple different types of gates.

4. What are the advantages of this enzymatic method compared to the strand displacement method? Since multiple enzymes are involved in this system, there would be gaps between this method and strand displacement method when considering the reaction speed and the cost.

The advantages of enzymatic methods come mostly from signal amplification and greater control. That comes with the tradeoff of more heat sensitivity and higher cost.

The Trumpet platform is indeed more complicated, and expensive, than strand displacement logic systems.

The presence of the enzymes is advantageous in that we have couple orders of magnitude of signal amplification, and more resistance to photobleaching. Transcribing the aptamer to obtain fluorescent readout, from each copy of DNA logic gate template we get few hundred copies of fluorescent aptamer (according to Milligan JF, Uhlenbeck, OC. *Methods Enzymol* (1989) 180:51-62).

Additionally, the actual fluorescence comes from the small molecule ligand of the aptamer, and the ligands are constantly exchanged between bound and unbound state. If a ligand molecule gets photobleached, there is a high likelihood of another, non-bleached ligand taking its place in the aptamer. This is an advantage over molecules with covalently bound fluorophore, where if a covalently attached fluorophore gets bleached the molecule will not regain fluorescence.

5. There is a typo in Figure 1b, the last two rows in the NAND truth table are the same.

Thank you for pointing it out. We have corrected the typo.

Reviewers' Comments:

Reviewer #1:

Remarks to the Author:

The authors have addressed all the comments raised by each of the reviewers. As such, the article can be accepted.

Reviewer #2:

Remarks to the Author:

The authors have addressed the concerns of this reviewer and a publication of the revised article is recommended.

We sincerely thank the Reviewers for all the constructive comments and suggestions that improved this article in all conceptual and experimental aspects.

Reviewer #1 (Remarks to the Author):

The authors have addressed all the comments raised by each of the reviewers. As such, the article can be accepted.

Thank you very much for the feedback.

Reviewer #2 (Remarks to the Author):

The authors have addressed the concerns of this reviewer and a publication of the revised article is recommended.

Thank you very much for the feedback.